# Highly Orientated Sericite Nanosheets in Epoxy Coating for Excellent Corrosion Protection of AZ31B Mg Alloy

**DOI:** 10.3390/nano13162310

**Published:** 2023-08-11

**Authors:** Hao Wu, Ke Xi, Yan Huang, Zena Zheng, Zhenghua Wu, Ruolin Liu, Chilou Zhou, Yao Xu, Hao Du, Yansheng Yin

**Affiliations:** 1Guangdong Key Laboratory of Materials and Equipment in Harsh Marine Environment, Guangzhou Maritime University, Guangzhou 510725, China; haowu.academic@gmail.com (H.W.); yanhuang1804@outlook.com (Y.H.); zhengzena0603@163.com (Z.Z.); wzh13423945193@163.com (Z.W.); rlliu@gzmtu.edu.cn (R.L.); duhao1972@126.com (H.D.); 2School of Naval Architecture and Ocean Engineering, Guangzhou Maritime University, Guangzhou 510725, China; 3School of Mechanical and Automobile Engineering, South China University of Technology, Guangzhou 510641, China; 201920101836@mail.scut.edu.cn; 4Guangdong Institute of Special Equipment Inspection and Research, Foshan 510655, China; xuyao092y@163.com

**Keywords:** magnesium, sericite, epoxy, corrosion, electrochemical

## Abstract

The growing demands for material longevity in marine environments necessitate the development of highly efficient, low-cost, and durable corrosion-protective coatings. Although magnesium alloys are widely used in the automotive and aerospace industries, severe corrosion issues still hinder their long-term service in naval architecture. In the present work, an epoxy composite coating containing sericite nanosheets is prepared on the AZ31B Mg alloy using a one-step electrophoretic deposition method to improve corrosion resistance. Due to the polyetherimide (PEI) modification, positively charged sericite nanosheets can be highly orientated in an epoxy coating under the influence of an electric field. The sericite-incorporated epoxy coating prepared in the emulsion with 4 wt.% sericite exhibits the highest corrosion resistance, with its corrosion current density being 6 orders of magnitude lower than that of the substrate. Electrochemical measurements and immersion tests showed that the highly orientated sericite nanosheets in the epoxy coating have an excellent barrier effect against corrosive media, thus significantly improving the long-term anti-corrosion performance of the epoxy coating. This work provides new insight into the design of lamellar filler/epoxy coatings with superior anticorrosion performance and shows promise in the corrosion protection of magnesium alloys.

## 1. Introduction

Global efforts are underway to reduce transportation-related greenhouse gas emissions, which can be achieved with the increased use of lightweight materials in the transportation industry [1,2]. Due to their low density, high elastic modulus, high specific strength, good heat conductivity, good shock resistance, and good castability, magnesium and its alloys are widely used as structural metal materials in terrestrial vehicles and aircrafts [3,4]. However, magnesium alloys experience severe corrosion issues in marine environments due to magnesium’s high chemical reactivity, which impedes their further application in naval architecture [5,6]. In recent decades, numerous efforts have been made to reduce magnesium’s susceptibility to corrosion. In addition to improving alloy composition and microstructure, coating technology is the most effective method for preventing and mitigating corrosion of Mg alloys [7,8]. Among anti-corrosion coatings, organic coatings such as polyurethane, epoxy resin, and acrylic resin have been widely used for the corrosion protection of magnesium alloys [9,10,11]. In addition to being environmentally friendly due to their low emission of volatile organic compounds, waterborne epoxy coatings offer superior chemical resistance and adhesion, making them ideal for the automotive and marine industries [11,12]. Nevertheless, during the drying process and film formation, gas bubbles and pinholes are unavoidably formed within the epoxy coating. These inherent defects in pure epoxy coatings allow H_2_O, O_2_, and ions to permeate into corrosive media, resulting in poor long-term corrosion resistance [13,14]. Consequently, many works have focused on enhancing the impermeability of organic coatings in order to improve their corrosion resistance [15].

To address the aforementioned issue, lamellar fillers (such as graphene, MXenes, and clays) with a high aspect ratio and good barrier capability were added to organic coatings [14,16,17,18]. For example, Yan et al. [14] prepared Ti_3_C_2_T_x_/epoxy composite coatings on aluminum alloy and achieved good anti-corrosion and wear-resistance properties. Zhu et al. [16] provided a facile strategy to design a composite coating of cationic dopamine-reduced graphene oxide (DRGO^+^) nanosheets and epoxy with good corrosion resistance. Hosseini et al. [17] also demonstrated that the addition of 5% polyaniline–montmorillonite clay into an epoxy coating on aluminum alloy can significantly improve corrosion resistance. However, lamellar materials, such as Ti_3_C_2_T_x_ and graphene, currently have drawbacks, such as difficult preparation and high cost, that limit their industrial applications on a large scale. Moreover, their potential electrical conductivity can result in a galvanic effect with the metal substrate, thereby increasing the likelihood of coating failure due to corrosion [19,20].

Natural clays have attracted interest in many fields due to their availability, abundance, environmental-friendliness, and low cost [21]. Sericite, a macroscopic transparent phyllosilicate clay, possesses impressive properties including high moduli, electric insulation, ultraviolet ray resistance, temperature stability, and chemical durability [22,23]. Sericite mineral raw materials can be processed into nanosheets or nanoplates using purification, intercalation, and exfoliation [24,25,26]. The high ratio of lateral to axial dimension makes them excellent lamellar fillers for use in coatings, as they provide an outstanding physical barrier. Our recent work [21] demonstrated that the addition of sericite nanosheets can significantly improve the corrosion resistance of micro-arc oxidation coating on aluminum alloy. According to the theoretic study by Lu and Mai [27], the geometric factors of clay fillers in coatings can significantly affect the barrier property of coatings. To minimize the permeability of a coating, the orientation of the clay fillers should be aligned parallel to the surface plane, and their size and aspect ratio should also be optimized. In addition, the bond strength at the filler/polymer interface can also significantly impact the structural stability of a composite coating [28]. Herein, we aim to fabricate a highly orientated sericite/polymer coating using the electrophoretic deposition of cationic-modified sericite nanosheets and waterborne epoxy. By intercalating cetyltrimethyl ammonium bromide (CTAB) and modifying sericite with branched polyethyleneimine (PEI), cationic-modified sericite nanosheets are successfully produced. The PEI-modified sericite (PEI-SER) nanosheets can be dispersed steadily in an aqueous solution and are used as a filler in a water-based cationic epoxy emulsion. In addition, the composite epoxy coating with aligned PEI-SER is fabricated on the surface of AZ31B magnesium alloy using electrophoretic deposition technology under the influence of an electric field (as shown in Figure 1). The anti-corrosion property of composite epoxy coatings incorporating sericite nanosheets is then evaluated, and the relevant mechanisms are also investigated and discussed.

## 2. Materials and Methods

### 2.1. Materials

Water-based cathodic epoxy resin electrophoresis emulsion (solid content: 35%) was supplied by Jiangxi Gaojie Technology Co., Ltd. (Taihe, China), and natural sericite (800 mesh) was obtained from Wanqiao Mica Inc. (Chuzhou, China). PEI (Mw = 25,000), CTAB, nitric acid (HNO_3_), silver nitrate (AgNO_3_), ethanol, and sodium chloride (NaCl) were purchased from Aladdin Industrial Corporation (Shanghai, China). AZ31B magnesium alloy specimens (40 × 25 × 2 mm) were used as the substrates.

### 2.2. Intercalation and Modification of Sericite

Natural sericite was intercalated using the following procedures. First, the natural sericite was heated at 800 °C in a muffle furnace for 1 h, and then the sericite powder was stirred with 5.0 mol/L HNO_3_ at 95 °C for 5 h, filtered and washed with deionized water until the pH was 7, and dried at 80 °C. Then, the powder was dispersed into a NaCl supersaturated solution at 95 °C, stirred for 3 h, and then filtered and washed with deionized water to remove redundant NaCl (tested using AgNO_3_). Subsequently, the resulting powder and a certain amount of CTAB (mass ratio: 15/46) were dispersed into deionized water at 80 °C, stirred for 24 h, filtered and washed with deionized water to remove redundant CTAB (tested using AgNO_3_), and then dried at 80 °C. For the sericite intercalated with CTAB, the product was named CTAB-SER.

After intercalation, a certain amount of CTAB-SER was dispersed into 500 mL ethanol and stirred in ultrasonication for 2 h. Then, the exfoliated CTAB-SER micro-plates were obtained using centrifuging and drying at 80 °C. Then, the exfoliated CTAB-SER nanosheets were modified with PEI using a typical method. Specifically, 5 g of CTAB-SER nanosheets, 100 mg of PEI, and 292 mg of NaCl were dissolved in 500 mL of deionized water, followed by stirring for 1 h. The resultant powder was centrifuged, washed with deionized water several times, and then dried at 80 °C for further use. For the sericite modified with PEI, the product was named PEI-SER.

### 2.3. Preparation of Composite Coatings

In this study, AZ31B magnesium alloy specimens (40 × 25 × 2 mm) were used as the substrates. They were ground and polished with 400, 1200, and 2000 grit silicon carbide paper, cleaned ultrasonically in ethanol, rinsed with distilled water, and dried in the air before coating.

Prior to electrophoretic deposition, the composite electrophoretic emulsion was prepared as follows. First, a given amount of modified sericite was ultrasonically dispersed in 10 g deionized water for 1 h to obtain a homogeneous suspension. Then, the homogeneous suspension was slowly added into 90.0 g of cathodic epoxy resin electrophoresis emulsion, and then the mixture was stirred under 3000 rpm for 30 min to achieve the uniform dispersion of sericite and epoxy. After stirring, it was dispersed using ultrasound for 1 h to achieve better dispersion and then degassed in a vacuum oven for 30 min to remove the air bubbles. For comparison, mixed emulsions with different modified sericite contents (0 wt.%, 1.0 wt.%, 2.0 wt.%, 4.0 wt.%, and 6.0 wt.%) were prepared using the same method.

Then, the electrophoretic deposition was conducted in a beaker with the platinum plate electrode as the anode, pretreated AZ31B magnesium alloy as the cathode, and the mixed electrophoretic emulsion placed in. The coating was deposited using electrophoresis at 60 V for 5 min. After coating, the sample was taken out of the electrophoretic emulsion, washed with deionized water, and dried at 150 °C for 30 min to obtain an epoxy composite coating. For convenience, the coatings obtained using emulsion deposition with different sericite contents were named E, E-S1, E-S2, E-S4, and E-S6, respectively.

### 2.4. Materials Characterization

Field-emission scanning electron microscopy (FE-SEM, SUPRA^®^ 55, Carl Zeiss, Oberkochen, Germany) was used to observe the surface morphology of sericite powders and the surface and cross-sectional morphologies of the coatings. The surface and lateral elemental distributions were determined using energy-dispersive X-ray spectroscopy (EDS). The X-ray diffraction patterns of untreated and as-treated sericite powders were characterized on an X-ray powder diffractometer (XRD, X’pert Powder, Malvern Panalytical, Malvern, UK) with Cu K_α_ radiation (λ = 0.15418 nm). The XRD data were collected in the 2*θ* range from 10° to 80° at a scanning rate of 12°/min and from 1° to 10° at a scanning rate of 3°/min. The Fourier transform infrared (FT-IR) spectra were obtained using an FT-IR spectrometer (Nicolet IS50, Thermo Fisher Scientific, Waltham, MA, USA) in the wavenumber range from 4000 to 500 cm^−1^ with a resolution of 1 cm^−1^. The composition of the coatings was determined using grazing-incidence X-ray diffraction (GIXRD, SmartLab, Rigaku, Tokyo, Japan) with Cu K_α_ radiation (λ = 0.15418 nm) at an incidence angle of 1°. The XRD data were collected in the 2*θ* range of 10°~90° at a scanning rate of 2°/min and step size of 0.02°.

### 2.5. Corrosion Evaluation

The corrosion characteristics of the samples were evaluated at room temperature in a 3.5 wt.% NaCl solution. The electrochemical tests were conducted on an electrochemical workstation (Reference 600+, Gamry, Warminster, PA, USA) using a three-electrode cell with the platinum mesh electrode as the counter electrode, the saturated calomel electrode (SCE) as the reference electrode, and the sample as the working electrode (the exposed area was 1 cm^2^). Electrochemical impedance spectroscopy (EIS) plots were obtained with an interference potential at 5 mV from 10^5^ Hz to 10^−1^ Hz. The equivalent circuits of EIS were fitted with Gamry Echem Analyst software. Potentiodynamic polarization (POL) was performed between −0.25 V and 0.5 V (vs. OCP) at a scanning rate of 1 mV/s. In order to obtain the polarization curves, the potential was swept from the cathodic to the anodic regions. Tafel extrapolation was used to derive the corresponding corrosion current densities (*i_corr_*) and corrosion potentials (*E_corr_*). All electrochemical tests were performed at least 3 times to ensure repeatability. Immersion tests were carried out to observe the corrosion behavior of the samples. After immersion in a 3.5 wt% NaCl solution at room temperature for 0, 1, 3, 5, 7, and 14 days, all samples were quickly rinsed and dried in the air. FE-SEM and 3D optical profilometry (UP Dual Model, Rtec, San Jose, CA, USA) were used to examine the surface and cross-sectional state of composite coatings after 14 days of immersion.

## 3. Results

### 3.1. Characterization of Sericite before and after Pretreatment

The wide-angle (Figure 2a) and the small-angle (Figure 2b) XRD spectra of the original and modified sericite reveal the evolution of interlaminar structure during the intercalation and modification process of sericite. The diffraction peaks in Figure 2a are consistent with muscovite (JCPDS 07-0032) and quartz (JCPDS 01-079-1910). After CTAB intercalation, the majority of peaks persist, indicating the absence of phase transformation and the preservation of the lamellar structure [22]. However, the intensity of peak *d*_002_ = 2.4 nm (2*θ* = 8°) is weaker than that of natural sericite, shifting to a lower diffraction angle than before CTAB intercalation. Concurrently, a new basal reflection corresponding to *d*_002_ = 5.2 nm (2*θ* = 2°) was observed, indicating that CTAB molecules enter the interlayer spaces in a regular arrangement and the interlamellar spacing is partly expanded [24,25,29]. After ultrasonication and PEI treatment, the intensity of *d*_002_ = 2.4 nm (2*θ* = 8°) further decreases and the broad peak *d*_002_ = 5.2 nm (2*θ* = 2°) disappears, suggesting further dissociation of the layered lattice along the vertical direction of the (002) plane. The XRD results indicate the layered structure of sericite is separated after pretreatment and modification.

FT-IR was used to characterize the functional groups of SER, CTAB, CTAB-SER, PEI, and PEI-SER. As depicted in Figure 2c, the 3600 cm^−1^ stretching band is attributed to the presence of –OH groups on the surface of sericite [26,30]. After CTAB intercalation, the peak in CTAB-SER at 3600 cm^−1^ is significantly diminished compared to SER. Furthermore, the spectrum of CTAB-SER displays two weak absorption bands at 2926 cm^−1^ and 2852 cm^−1^, which can be attributed to the asymmetric and symmetric stretching vibrations of methylene (-CH_2_) [31]. The Si–O stretching vibration band (at 1077 cm^−1^) in CTAB-SER is broader than that of SER, which can be due to the appearance of the C–N and C–C bending vibration bands at 960 cm^−1^ and 909 cm^−1^, respectively [25]. According to the above XRD and FT-IR results, it can be concluded that the intercalation was successful because CTA^+^ entered the interlayer space of sericite. The characteristic peaks in PEI appear at ~3300 cm^−1^ (-NH and -NH_2_ stretching) and 1630 cm^−1^ (-NH_2_ scissoring vibration) [32]. The typical peaks in CTAB-SER are still present in the PEI-SER spectrum, and two new peaks appear at 3355 cm^−1^ and 1630 cm^−1^. The FT-IR results confirm that the PEI is successfully attached to the surface of the sericite nanosheets.

Figure 2d,e depicts the zeta potential and size distribution of SER, CTAB-SER, and PEI-SER dispersed in deionized water. PEI-SER appears to be positively charged with a zeta potential of 45.3 mV, whereas SER and CTAB-SER exhibit negative zeta potentials of 23.6 mV and weak positive zeta potentials of 8.9 mV, respectively. As a result of their positive charge, the sericite nanosheets move toward the cathode during electrophoretic deposition in order to form a composite coating. Although the measured particle size is not the actual lamellar diameter but the equivalent particle size due to sericite’s two-dimensional layered structure, it can still approximate the lamellae size. The measured particle size of raw SER ranges from approximately 2000 to 7000 nm, with an average size of 3585.9 nm. CTAB-SER and PEI-SER display a sharper Gaussian-like size distribution than raw SER, with an average size of 1990.2 nm and 1718.2 nm, respectively. Figure 2f compares SER and PEI-SER at a concentration of 0.5% by weight dispersed in deionized water after one hour of standing. Due to the agglomeration of sericite, the natural sericite solution exhibits distinct stratification and precipitation after 1 h of standing, whereas the PEI-SER solution does not exhibit any discernible precipitation. The surface morphology of SER and PEI-SER were observed using SEM. Figure 2g demonstrates that raw sericite possesses a typical compact and layered structure, whereas PEI-SER exhibits a structure with thinner layers. The PEI-SER lamellae were obtained with exfoliation and stripping from the multi-layered CTAB-SER using ultrasonic and stirring treatment. Consequently, based on the aforementioned results, it can be concluded that CTAB intercalation dissociated the multilayer structure of the original sericite, whereas PEI modification successfully modulated the surface charge properties of the sericite nanosheets.

### 3.2. Surface Morphology and Cross-Section of Coatings

Figure 3 depicts the surface morphology and cross-sections of coating surfaces prepared with emulsions containing varying amounts of PEI-SER. The sericite nanosheets are nearly uniformly dispersed in the coatings and closely combined with the epoxy resin, demonstrating excellent interfacial compatibility. As the sericite concentration in the electrophoretic emulsion increases, the density of sericite nanosheets observed from the surface SEM image and the Si element content of the coating also increase (Appendix A). The average thickness of the pure epoxy coating is approximately 26 μm, whereas the average thickness of four coatings with sericite is approximately 30 μm, indicating that the addition of PEI-SER increases the electrophoretic deposition rate of epoxy coating. This can be attributed to the molecular brush effect of positively charged sericite, which facilitates the movement of the emulsion toward the cathode during electrophoresis, resulting in denser and thicker coatings [16]. The majority of sericite nanosheets are aligned parallel to the sample surface in the PEI-SER coating. Throughout the electrophoretic deposition, a perpendicular DC electric field is applied to the metal surface. Therefore, the positively charged PEI-SER nanosheets are propelled toward the cathode by the electric field. Due to the uniformly applied electric field force and electrostatic repulsion force between nanosheets, the nanosheets tend to align themselves parallel to the surface of the sample during movement. As shown in Figure 3g,k,o, sericite nanosheets are uniformly dispersed in parallel in the E-S1, E-S2, and E-S4 coatings. In addition, the sericite nanosheets are compatible with the epoxy matrix at the interface. The parallel arrangement of sericite nanosheets in epoxy coatings becomes more pronounced and denser as sericite content is continuously added. The parallel distribution and high density of sericite nanosheets are advantageous for enhancing the coating’s physical barrier effect and resistance to corrosion. However, when the concentration of PEI-SER nanosheets reaches a certain threshold, disordered arrangement and agglomeration begin to emerge. As seen in Figure 3s, the sericite nanosheets in the E-S6 coating have become disordered and agglomerated, which may lead to the formation of micropores and defects that diminish the impermeability of composite coatings.

### 3.3. Electrochemical Corrosion Tests

Figure 4a depicts the potentiodynamic polarization (POL) curves for the different samples after immersion in 3.5 wt.% NaCl for 1 h. The corrosion potentials (*E_corr_*) and corrosion current densities (*i_corr_*) obtained directly from the Tafel region in the cathodic polarization curves using Tafel extrapolation are listed in Table 1 and shown in Figure 4b, respectively. As the concentration of sericite increased, the polarization curve gradually shifted to the left and upward in the order of ES-1, ES-2, ES-6, and ES-4, with E-S4 occupying the most extreme left and highest position on the POL curves plot. The relatively lower corrosion potential of E, ES-1, and ES-2 can be due to the galvanic effect between the exposed area and the coated area on the substrate due to local defects. Among all samples, E-S4 shows the smallest *i_corr_* of 2.412 *×* 10^−11^ A·cm^−2^, which is approximately 2 orders of magnitude lower than that of E and 6 orders of magnitude lower than the AZ31B Mg substrate. Moreover, E-S4 has the highest *E_corr_* of −1.214 V. More positive Ecorr and lower *i_corr_* are generally indicative of greater corrosion resistance [13]. Consequently, the anti-corrosion performance of epoxy samples is significantly enhanced with the addition of sericite. This can be attributed to the self-aligned parallel arrangement of sericite nanosheets, which can significantly affect the barrier performance of the epoxy coating and enhance its corrosion resistance. In comparison with E-S4, E-S6 demonstrates a lower *E_corr_* and a higher *i_corr_*, indicating that the disordered arrangement and aggregation of sericite nanosheets negatively impacted the coating’s corrosion resistance performance. Nevertheless, due to its relatively high sericite concentration, its performance is still superior to all other samples except E-S4.

Figure 5a–c depicts the Nyquist, Bode-impedance, and Bode-phase plots for the samples after immersion for 1 h. For the Nyquist curves, a larger diameter of capacitive loops indicates a coating with superior anti-corrosion properties [13]. Notably, as shown in Figure 5a, the capacitive loops enlarge after electrophoretic deposition, indicating improved corrosion resistance compared with the substrate. In the case of the Bode impedance plots, the impedance modulus at the lower frequency (|*Z*|_0_._1Hz_) is frequently utilized as a crucial indicator for evaluating the anticorrosion performance of coatings. Usually, a higher |*Z*|_0_._1Hz_ represents better corrosion protection [33,34]. The bode impedance plot in Figure 5b demonstrates that the E-S4 has the highest |*Z*|_0_._1Hz_ value, suggesting better corrosion protection performance than other samples.

According to the characteristics of the Bode and Nyquist plots, equivalent circuits (Figure 5d) are used to fit the EIS data from the different samples. In this work, the constant phase element (CPE) is used to represent the non-ideal capacitors, which is expressed by:Y = Y_0_ (jω)^n^,(1)
where Y_0_ and n are the admittance constant and empirical exponent, respectively [35,36]. Based on the characteristics of the EIS data and previous studies [4,37], R_s_(CPE_f_ (R_pore_(((CPE_dl_R_ct_)(CPE_diff_R_diff_))))) is proposed to fit the EIS data for E-S1, E-S2, and E-S6. The EIS data curves for AZ31B show obvious inductive characteristics in the low-frequency impedance region, which can be attributable to pit formation during corrosion [38]. Consequently, R_s_(((CPE_dl_R_ct_)(CPE_diff_R_diff_))(LR_L_)))) is proposed to fit the EIS data for AZ31B. Since the EIS data curve for E-S4 does not exhibit the full low-frequency diffusion impedance region, R_s_(CPE_f_(R_pore_(CPE_dl_R_ct_))) is proposed to fit E-S4. In these three circuits, R_s_ refers to the solution resistance. CPE_f_ represents the capacitance of the coating, and R_pore_ stands for the total resistance of pores and defects in the coating. CPE_dl_ represents the capacitance of the electric double layer at the low-frequency area, and R_ct_ refers to the charge transfer resistance in the Faradic process. CPE_diff_ represents the capacitance pertaining to the diffusion process, and R_diff_ denotes the relevant resistance. L refers to the inductance associated with the relaxation process of adsorbed species, and R_L_ is the relevant inductance resistance. The fitted data are shown in Table 2. Generally, the capacitive curves at the high-frequency region stand for the surface film and charge transfer, while lower-frequency behavior is associated with mass transfer [38,39,40]. For these epoxy-coated samples, the R_pore_ can be used to evaluate the barrier effect of coating and the R_ct_ can be used to evaluate the ability of coating to maintain the coating/metal interface corrosion [41]. Both of these two parameters indicate a general trend in reducing corrosion resistance: E-S4 > E-S6 > E-S2 > E-S1 > E.

In order to evaluate the long-term corrosion behavior and mechanism, EIS is utilized after exposing samples to 3.5 wt. % NaCl for varying time periods. The EIS data are represented in Figure 6 as Nyquist, Bode-impedance, and Bode-phase plots. The appearance of an inductive loop can be used to determine the onset of localized corrosion. Figure 7 depicts the variation between the coating impedance (*R_pore_*) and the impedance modulus at low frequencies (|*Z*|_0_._1Hz_). The *R_pore_* and |*Z*|_0_._1Hz_ of the samples decreases as immersion time increases, indicating that the barrier capability of coatings and overall corrosion resistance of the samples would gradually decrease during immersion. The addition of sericite effectively slows the deterioration of the coating’s corrosion resistance, and the E-S4 sample demonstrates the best long-term corrosion resistance.

### 3.4. Immersion Tests

The immersion test was carried out for 14 days in 3.5 wt.% NaCl to investigate the corrosion process. The digital pictures of immersed specimens are shown in Figure 8. After 1 day of immersion, the AZ31B substrate begins to show signs of discoloration, filiform corrosion, and cracking. The corrosion gradually intensifies and spreads across the entire surface as the immersion time increases. For the sample with an epoxy coating, corrosion is significantly inhibited. However, due to flaws and pinholes in the coating, corrosive media can easily penetrate the pure epoxy coating. After 1 day of immersion, some tiny blisters and pores begin to appear on the surface of the E sample. With prolonged immersion, surface blistering and localized corrosion in edge regions continue to increase. The corrosion damage of samples with sericite is less severe than that of sample E. Preliminary coating damage starts on the 3rd day of E-S1 and E-S6, and the 5th day of E-S2. In contrast, the corroded areas of these samples are significantly smaller than those of sample E. Notably, after 14 days, only a few bubbles and edge corrosion areas appeared on the surface of E-S4, indicating that the coating still retains good corrosion resistance. CLSM and 3D topography images are used to observe the microscopic corrosion of samples after immersion for 14 days. As shown in Figure 9, the AZ31B substrate and E exhibit a surface that is heavily corroded and covered with corrosion products. Comparatively, the surfaces of samples containing sericite are considerably more intact and refined, indicating a milder corrosion behavior. Nevertheless, after 14 days of immersion, these samples still display shallow pits and corrosion products. Among them, the E-S4 sample has the lowest R_a_, indicating that the E-S4 coating has the most effective corrosion barrier effect.

### 3.5. Corrosion Products Analysis

The GIXRD patterns of all samples in 3.5 wt.% NaCl solutions after immersion for 14 days are shown in Figure 10. In the sericite/epoxy composite coatings, the broad peak of 2*θ* between 10° and 30° indicates epoxy, while the diffraction peaks at 17.5°, 26.6°, and 27.8° provide evidence of sericite (JSPDS NO. 47-1144). Corrosion products such as Mg(OH)_2_ (Brucite, JCPDS NO. 82-2455), MgO (JCPDS NO. 89-7746), MgAl-LDH (Hydrotalcite, JCPDS NO. 89-0460), and MgCl_2_ (JCPDS NO. 80-1752) can also be observed as diffraction peaks on the surface of the sample. Furthermore, the E sample has the highest relative intensity of the Mg (JCPDS NO. 89-7195) peak at 34.4° compared with the epoxy peak, indicating that the E sample has the lowest coating integrity and the largest area of substrate exposure.

SEM and EDS were used to determine the corrosion morphology and elemental composition of the samples following an immersion test. Figure 11 shows typical regions of samples after immersion in 3.5 wt.% NaCl for 14 days. As shown in Figure 11a,b, the surface of AZ31B is fully covered with cotton-like corrosion products. The elemental composition of the corroded area indicates that the corrosion products primarily consist of MgO and Mg(OH)_2_, with a small amount of MgAl-LDH and chlorides [42]. On the surface of the E sample, many cracks are distributed, and the coating is partially peeled off. On the exposed substrate, corrosion products can be observed in continuous quantities. This is due to the fact that when the corrosive medium penetrates into the coating, the substrate will be severely corroded to produce hydrogen gas and corrosion products, resulting in local cracks and defects in the coating [43,44]. Sericite nanosheets can be observed from the surfaces of sericite-containing samples, with the Si element also detected in related areas. Instead of cracks and massive corrosion products, these sericite-containing samples typically exhibit corrosion pits and sheet-like corrosion products. The surface corrosion products primarily consist of sheet-like Mg(OH)_2_ and Mg-Al LDH, which are formed by continuous exudation of Mg and Al ions from the AZ31B substrate [37].

To analyze the internal condition of coatings after 14 days of immersion in 3.5% NaCl, cross-sectional SEM images and EDS maps were acquired. As shown in Figure 12a, the distribution of C and O elements reveals that after immersion, the epoxy coating on E has been replaced with a thick layer of corrosion products. The coatings of E-S1 and E-S2 are also severely corroded, as evidenced by the presence of deep cracks within and beneath the epoxy coating. The distribution of Mg in Figure 12b,c also indicates the formation of corrosion products on and in these coatings. In contrast, Figure 12d demonstrates that the E-S4 coating is relatively intact after 14 days of immersion. It is almost difficult to observe the aggregation of Mg elements on its surface, indicating that fewer corrosion products are generated. However, the SEM image and EDS maps in Figure 12e indicate that the agglomeration of sericite in the E-S6 coating may cause localized defects in the surrounding area and ultimately lead to corrosion failure.

## 4. Discussion

In order to discuss the anticorrosion mechanism underlying the highly oriented epoxy-sericite composite coating, the electrochemical or chemical reaction changes at the substrate-coating interface were analyzed. The main anodic and cathodic reactions occurring at the interface during the electrochemical corrosion of magnesium and magnesium alloys are as follows:

Anodic dissolution:M → M^x+^ + xe^−^(2)

The M in Equation (2) is mainly Mg and Al for the AZ31 alloy. The cathodic reaction can be an oxygen reduction reaction (ORR) or hydrogen evolution reaction (HER):O_2_ + 2H_2_O + 4e^−^ → 4OH^−^(3)
2H_2_O + 2e^−^ → H_2_ + 2OH^−^(4)

The dissolved metallic ions in (2) can diffuse and react with OH^−^ to form corrosion products such as Mg(OH)_2_, MgO, and Mg-Al LDH:Mg^2+^ + 2OH^−^ → Mg(OH)_2_(5)
Mg(OH)_2_ → MgO + H_2_O(6)
Mg^2+^ + Al^3+^ + OH^−^ → Mg_x_Al_y_(OH)_z_(7)

These corrosion products can form at or around the active site where corrosion occurs. Due to the Pilling–Bedworth ratio (PBR) of Mg, these corrosion products may form defective passivation layers and provide partial protection. However, in chloride solutions, Cl^−^ can destroy the passivation layer and accelerate the corrosion of magnesium, which can be expressed by (8) and (9):Mg^2+^ + 2Cl^−^ → MgCl_2_(8)
MgCl_2_ + 2OH^−^ → Mg(OH)_2_ + 2Cl^−^(9)

Therefore, it can be inferred that the availability of H_2_O, O_2_, and Cl^−^ at the interface are the critical factors affecting the entire corrosion process.

As can be seen from the SEM images in Figure 11 and Figure 12, the corrosion failure mode of epoxy coatings on magnesium alloys can be divided into several stages. In the early stage of corrosion of the E sample, the epoxy coating possesses a corrosion retardation effect by preventing the corrosive species from the substrate. However, as the immersion time increases, the corrosive medium can diffuse through the coating via micro-pores and defects to reach the substrate/coating interface (Figure 13a). The corrosion of the Mg alloy substrate at the substrate/coating interface causes the formation of corrosion products (e.g., Mg(OH)_2_ and MgO) and the evolution of H_2_, leading to the accumulation of corrosion products and gas blisters under the coating and gradually resulting in the damage of coating integrity (Figure 13b). Furthermore, Cl^−^ can migrate through the coating and oxide, playing a crucial role in the dissolution of passivating hydroxide or oxide of magnesium and causing pitting corrosion. Due to the continuous erosion of Cl^−^, local corrosion such as pitting and blistering under the coating will be aggravated, and finally, the epoxy coating will collapse, as shown in Figure 13c.

According to the aforementioned corrosion mechanism and anti-corrosion mechanism for 2D layered materials in coatings [14,16,17], the anti-corrosion mechanism underlying sericite nanosheets in uniform orientation sericite/epoxy coating is discussed, as conceptually illustrated in Figure 13d. Under the influence of an electric field, an appropriate amount of sericite will be aligned in the coating, which can form a tight barrier network against diffusion of O_2_, H_2_O, and Cl^−^. For the substrate protected with a coating containing sericite, the low availability of O_2_ and H_2_O suppresses the cathodic reactions, which in turn suppresses the anodic reaction of Mg dissolution. In addition, the corrosion rate of the substrate is also reduced because fewer Cl^−^ ions pass through the coating. Generally, the addition of more layered fillers can increase the density of mico-/nano-plates in the coating, leaving fewer penetration paths for corrosive species, thereby further improving the impermeability of the coating [45]. Consequently, our study reveals that the corrosion resistance of E-S4 and E-S6 is superior to that of samples containing a lower concentration of sericite. However, when the concentration of sericite reaches a certain threshold, the dispersion and orderliness of nanosheets in epoxy begin to diminish. The aggregation of nanosheets within a coating can result in defects in the surrounding area, allowing corrosive media to penetrate. This may help explain why the anticorrosion performance of E-S6 coatings is inferior to that of E-S4 coatings.

## 5. Conclusions

In summary, CTAB-intercalated and PEI-modified sericite nanosheets were used to prepare a highly orientated sericite/epoxy coating on AZ31 Mg alloy using electrophoretic deposition. The anticorrosion performance of sericite/epoxy composite coatings was evaluated, and the results indicated that the highly orientated sericite nanosheets significantly improved the corrosion resistance of epoxy coatings. The potentiodynamic polarization test reveals that the E-S4 coating had the smallest i_corr_, which was approximately two orders of magnitude lower than the pure epoxy coating and six orders of magnitude lower than the AZ31B Mg substrate. The long-term immersion test demonstrated that that the E-S4 coating can still effectively protect the substrate from severe corrosion after 14 days of immersion in 3.5 wt.% NaCl solution. The increased corrosion resistance is attributed to the barrier effects of the highly orientated sericite nanosheets, which significantly delay the penetration of corrosive media. The enhanced corrosion resistance stems from the barrier effects of the sericite nanosheets arranged in parallel, which significantly retards the intrusion of corrosive media. This work provides insights into the design of high-performance and low-cost electrophoretic anticorrosion coatings, which have great application potential in the corrosion protection of metals.

## Figures and Tables

**Figure 1 nanomaterials-13-02310-f001:**
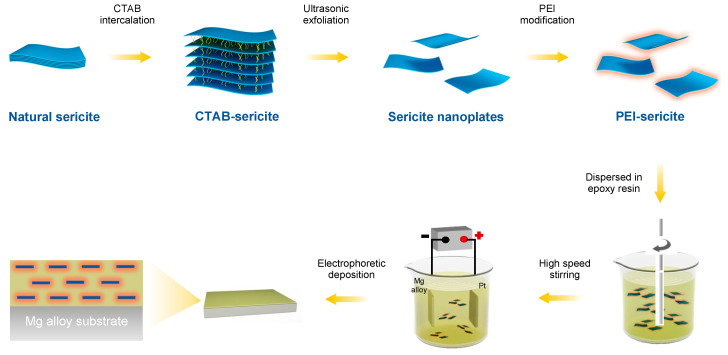
A schematic showing the fabricating procedure for highly orientated sericite/epoxy coating on AZ31B Mg alloy.

**Figure 2 nanomaterials-13-02310-f002:**
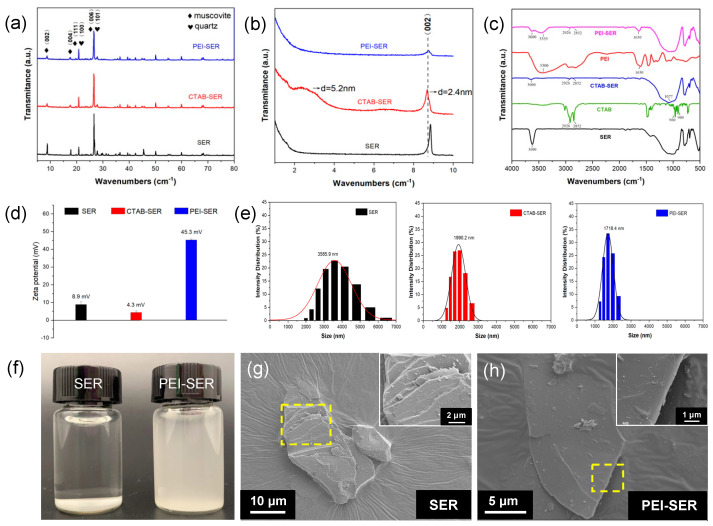
(**a**,**b**) XRD spectra of as-received and modified sericite. (**c**) FT-IR spectra of SER, CTAB, PEI, and PEI-SER. (**d**,**e**) SEM micrographs of SER (**d**) and PEI-SER (**e**), where the insets depict the magnified images of the selected area. (**f**) SER and PEI-SER solution with a concentration of 0.5 mg/mL after standing for 1 h. (**g**,**h**) Zeta potential distribution (**g**) and particle size distribution (**h**) of SER, CTAB-SER, and PEI-SER.

**Figure 3 nanomaterials-13-02310-f003:**
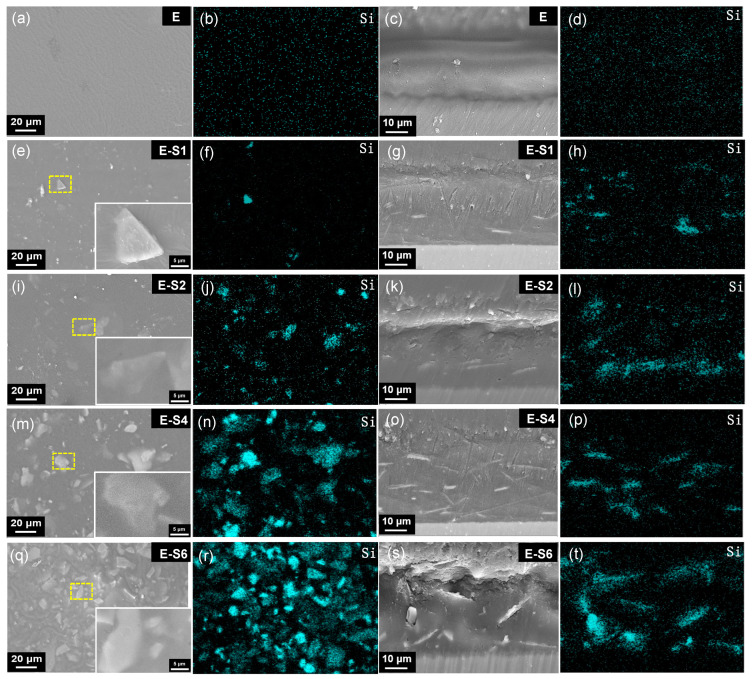
SEM images and Si elemental EDS maps showing the surface and cross-sections of different samples: E (**a**–**d**), E-S1 (**e**–**h**), E-S2 (**i**–**l**), E-S4 (**m**–**p**), and E-S6 (**q**–**t**). The first to the fifth rows are related to samples E, E-S1, E-S2, E-S4, and E-S6, respectively. The first column of images depicts the surface morphology of different samples with insets showing a magnified selected area, and the second column shows the Si elemental EDS maps of the related whole surface area of the first column. The third column of images depicts the cross-sections of different samples, and the fourth column shows the Si elemental of the related cross-sectional area of the third column.

**Figure 4 nanomaterials-13-02310-f004:**
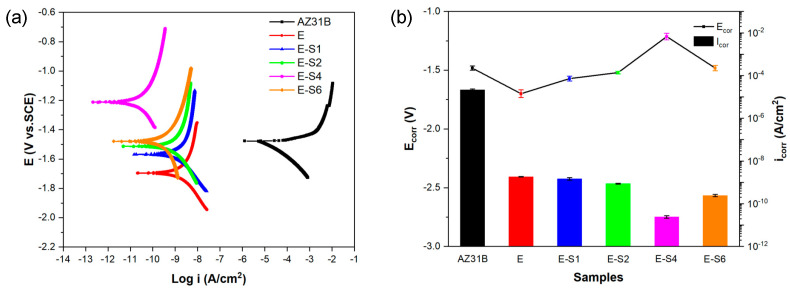
(**a**) Potentiodynamic polarization curves for the AZ31B alloy and coating samples after immersion in 3.5 wt.% NaCl solution for 1 h. (**b**) Comparison of the corrosion potentials (*E_corr_*) and corrosion current densities (*I_corr_*) calculated from the polarization curves.

**Figure 5 nanomaterials-13-02310-f005:**
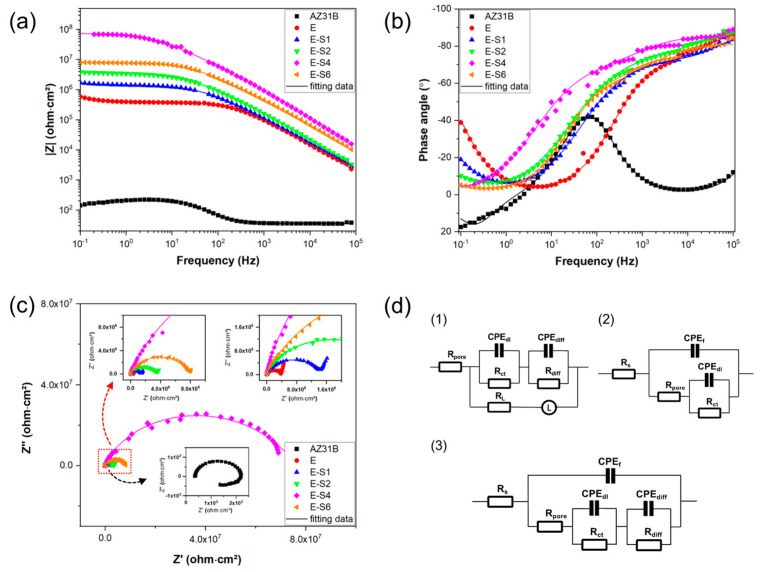
(**a***–***c**) Nyquist, Bode-impedance, and Bode-phase plots for the AZ31B Mg substrate, E, E-S1, E-S2, E-S4, and E-S6 after immersion in 3.5 wt.% NaCl for 1 h. (**d**) equivalent circuits.

**Figure 6 nanomaterials-13-02310-f006:**
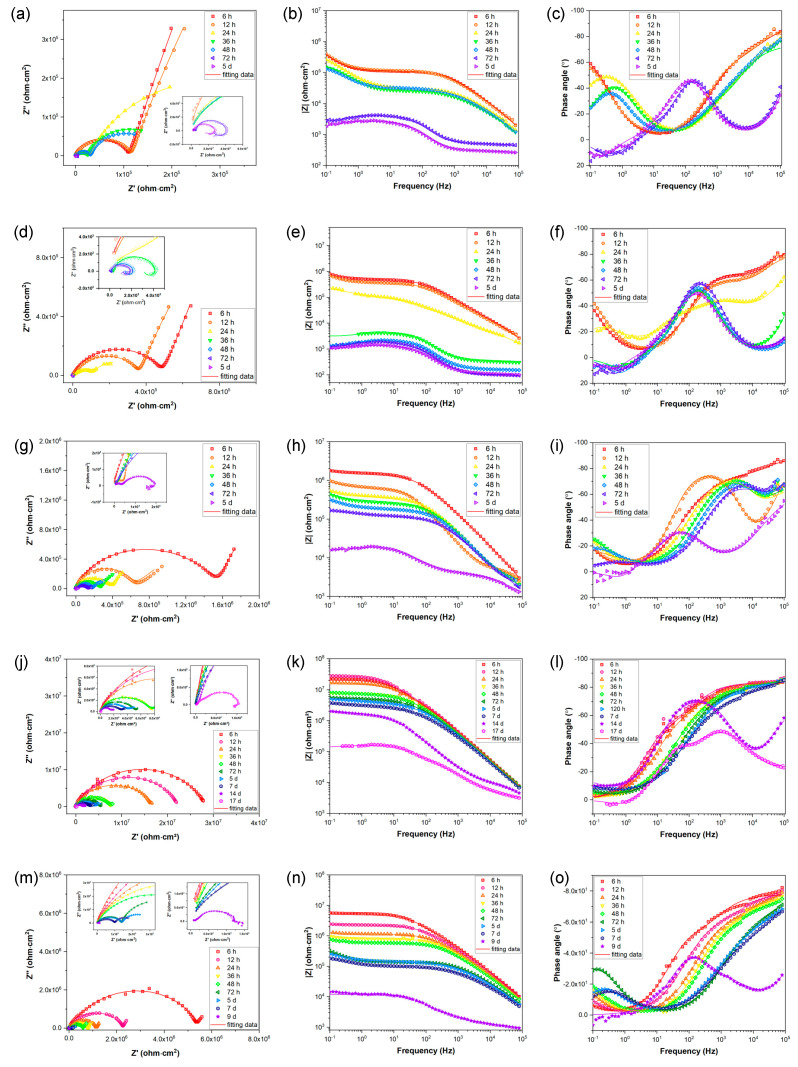
(**a**,**d**,**g**,**j**,**m**) Nyquist plots, (**b**,**e**,**h**,**k**,**n**) Bode-impedance plots, and (**c**,**f**,**i**,**l**,**o**) Bode-phase angle plots for (**a***–***c**) E, (**d***–***f**) E-S1, (**g***–***i**) E-S2, (**j***–***l**) E-S4, and (**m***–***o**) E-S6 after immersion for different times.

**Figure 7 nanomaterials-13-02310-f007:**
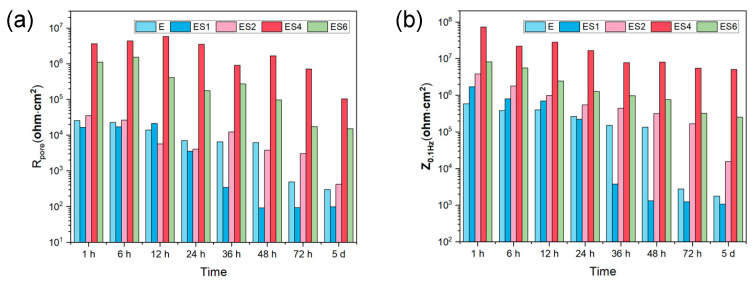
The change in *R_pore_* and |*Z*|_0_._1Hz_ against different immersion times from 1 h to 5 d.

**Figure 8 nanomaterials-13-02310-f008:**
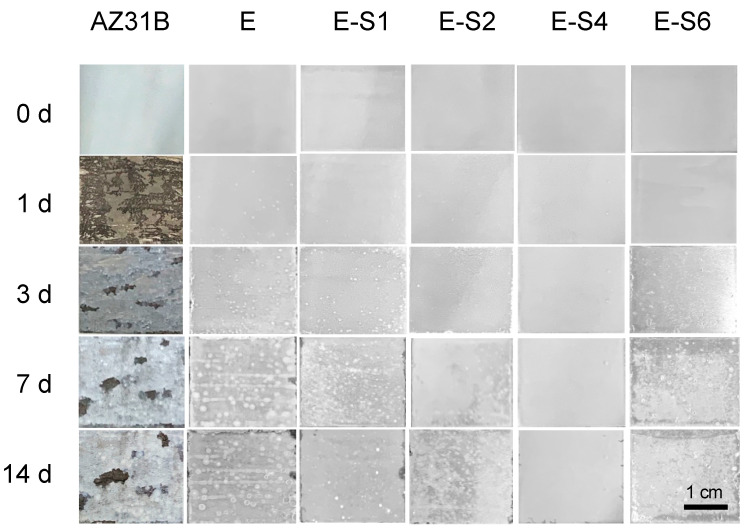
Digital pictures showing the AZ31B substrate, E, E-S1, E-S2, E-S4, and E-S6 after immersion for different times.

**Figure 9 nanomaterials-13-02310-f009:**
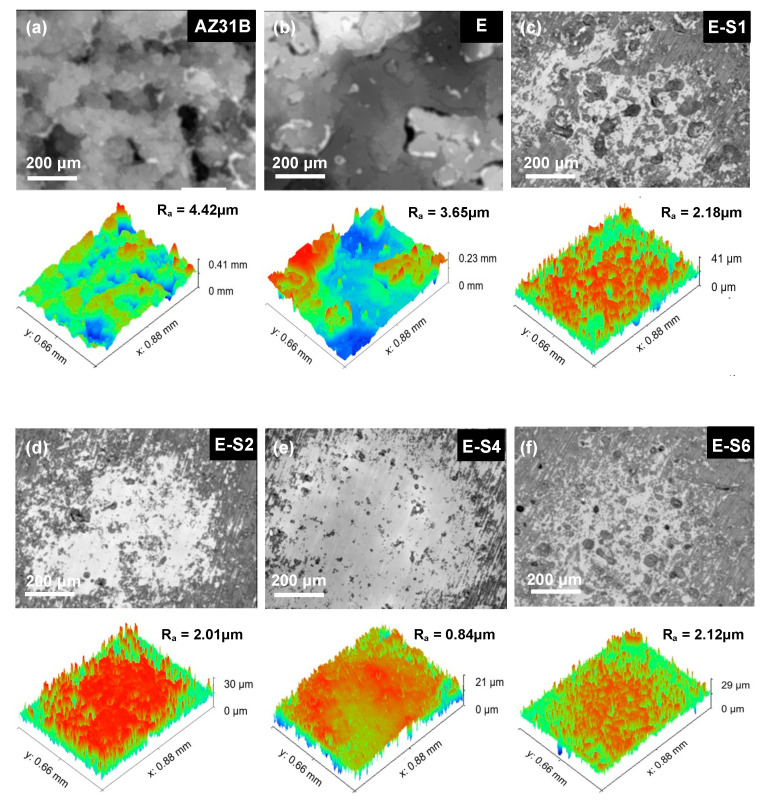
Confocal laser scanning microscopy (CLSM) and 3D topography images of samples after immersion in 3.5 wt.% NaCl for 14 days: (**a**) AZ31B; (**b**) E; (**c**) E-S1; (**d**) E-S2; (**e**) E-S4; (**f**) E-S6.

**Figure 10 nanomaterials-13-02310-f010:**
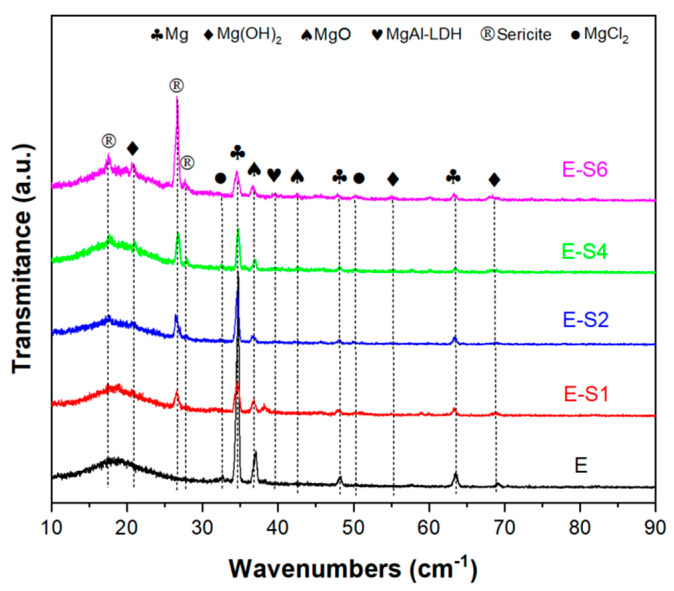
GIXRD patterns of samples after immersion in 3.5 wt% NaCl for 14 days.

**Figure 11 nanomaterials-13-02310-f011:**
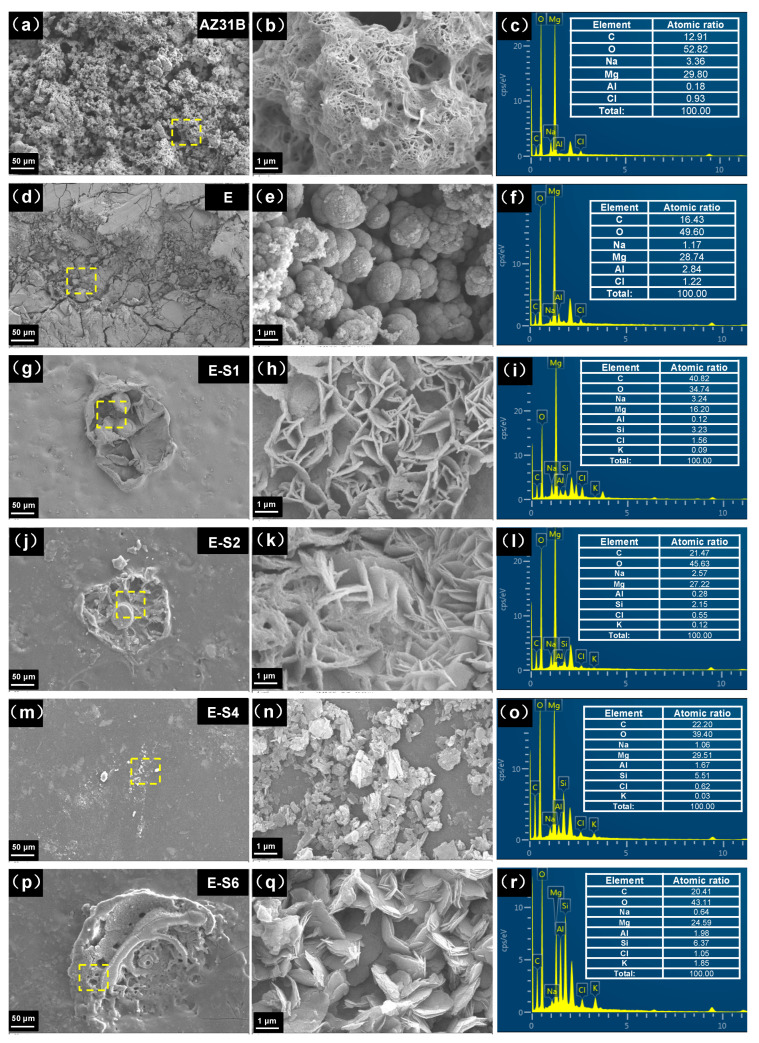
SEM images and corresponding EDS results of (**a**–**c**) AZ31B, (**d**–**f**) E, (**g**–**i**) E-S1, (**j**–**l**) E-S2, (**m**–**o**) E-S4, and (**p**–**r**) E-S6 after immersion in 3.5 wt% NaCl of 14 days.

**Figure 12 nanomaterials-13-02310-f012:**
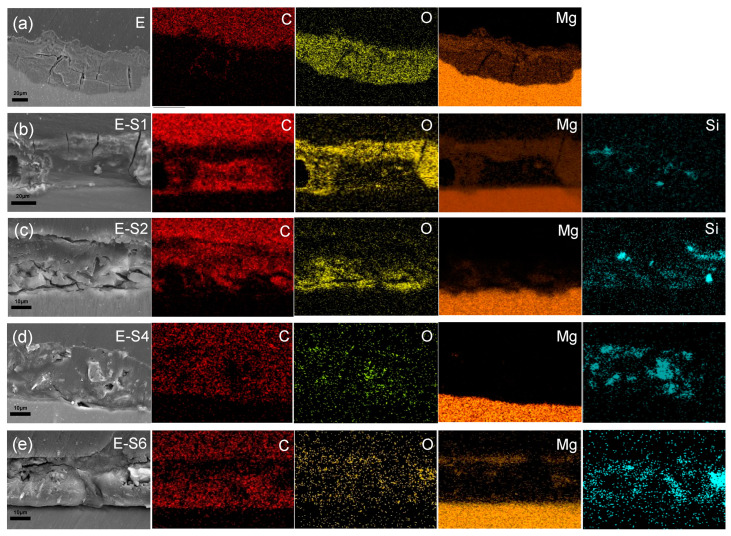
Cross-sectional SEM images and corresponding EDS mapping for (**a**) E, (**b**) E-S1, (**c**) E-S2, (**d**) E-S4, and (**e**) E-S6 after immersion in 3.5 wt% NaCl for 14 days.

**Figure 13 nanomaterials-13-02310-f013:**
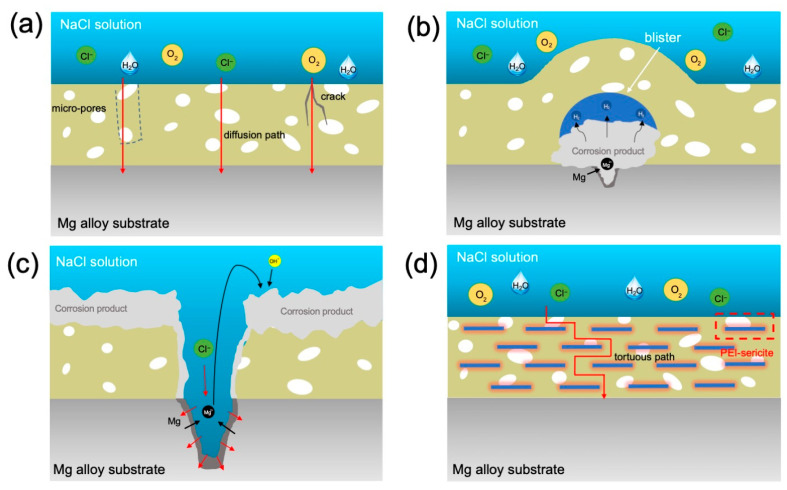
Schematic diagram showing the anti-corrosion mechanism: (**a**–**c**) different stages of the corrosion process for a bare epoxy coating and (**d**) the anti-corrosion mechanism underlying the uniform-oriented sericite nanosheets in epoxy coating.

**Table 1 nanomaterials-13-02310-t001:** *E_corr_*, *i_corr_*, and *β_c_* of different samples in 3.5 wt% NaCl calculated from the polarization curves.

Sample	*E_corr_* (V vs. SCE)	*i_corr_* (A·cm^−2^)	*β_c_* (V/decade)
AZ31B	−1.483 ± 0.018	(2.153 ± 0.19) × 10^−5^	−0.138 ± 0.006
E	−1.700 ± 0.032	(1.831 ± 0.05) × 10^−9^	−0.216 ± 0.009
E-S1	−1.571 ± 0.021	(1.464 ± 0.20) × 10^−9^	−0.196 ± 0.008
E-S2	−1.520 ± 0.011	(8.723 ± 0.43) × 10^−10^	−0.238 ± 0.001
E-S4	−1.214 ± 0.028	(2.412 ± 0.35) × 10^−11^	−0.260 ± 0.002
E-S6	−1.482 ± 0.023	(2.435 ± 0.29) × 10^−10^	−0.247 ± 0.002

**Table 2 nanomaterials-13-02310-t002:** Fitted EIS results for different samples after immersion in 3.5 wt% NaCl solution for 0.5 h based on the corresponding equivalent circuit models.

	AZ31B	E	E-S1	E-S2	E-S4	E-S6
Equivalent Circuit	R(((QR)(QR))RL)	R(Q(R((QR)(QR))))	R(Q(R((QR)(QR))))	R(Q(R((QR)(QR))))	R(Q(R(QR)))	R(Q(R((QR)(QR))))
*R_s_* (ohm·cm^2^)	15.60	11.88	12.95	10.46	14.60	4.88
*Y*_0_*_f_* (ohm^−2^·cm^−2^·S^−n^)	-	9.72 × 10^−10^	6.36 × 10^−10^	5.94 × 10^−10^	2.35 × 10^−10^	6.73 × 10^−10^
*n_f_*	-	0.9702	1	1	0.9529	0.9040
*R_pore_* (ohm·cm^2^)	-	1.57 × 10^4^	1.64 × 10^4^	3.56 × 10^4^	3.64 × 10^6^	1.12 × 10^6^
*Y*_0_*_dl_* (ohm^−2^·cm^−2^·S^−n^)	2.78 × 10^−8^	1.29 × 10^−8^	1.26 × 10^−8^	1.00 × 10^−8^	1.35 × 10^−9^	2.13 × 10^−9^
*n_dl_*	1	0.6781	0.6937	0.6506	0.6338	0.7179
*R_ct_* (ohm·cm^2^)	20.23	1.55 × 10^6^	2.81 × 10^6^	3.61 × 10^6^	7.27 × 10^7^	6.83 × 10^6^
*Y*_0_*_diff_* (ohm^−2^·cm^−2^·S^−n^)	4.65 × 10^−5^	2.61 × 10^−6^	2.654 × 10^−6^	2.417 × 10^−6^	-	2.29 × 10^−6^
*n_diff_*	0.9449	0.9553	0.8420	0.8556	-	0.8048
*R_diff_* (ohm·cm^2^)	179.6	2.48 × 10^6^	9 × 10^7^	2.13 × 10^7^	-	1.32 × 10^7^
*L* (H)	252.2	-	-	-	-	-
*R_L_* (ohm·cm^2^)	223.1	-	-	-	-	-
χ^2^	1.39 × 10^−3^	7.86 × 10^−4^	6.74 × 10^−4^	7.48 × 10^−4^	1.67 × 10^−3^	5.34 × 10^−4^

## Data Availability

Not applicable.

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
