# Peer review of "Highly Orientated Sericite Nanosheets in Epoxy Coating for Excellent Corrosion Protection of AZ31B Mg Alloy"

_nanomaterials, 2023, doi:10.3390/nano13162310_

Round 1
Reviewer 1 Report
The authors present a protection material against AZ31B alloy corrosion based on sercitite/epoxy
The work is intresting in the context of developing new materials against corrosion effect.
Comments:
1. Abstract: please provide meaning of PEI
2. Abstract: more scientific data are required
3. Figures 2 c, d, e: the text (data) from figures should be more clear (a higher resolution or higher text fonts is required)
4. Figure 2 (SEM) please rename figure 2 with figure 3. Also, figure 3 show a lack of clarity (to many information). I would suggest that the pictures corresponding to the mapping of elements and their table to be moved at supplementary informations
5. Please provide an explanation for the corrosion potential evolution (for instance why Ecor for Ma alloy is more positively when compared with E, E-S1, and E-S2)
6. Figure 5d: higher resolution of the text is required
7. Figure 6 should be moved to supplementary information (small information provided in manuscript with regard to this figure) with a higher resolution
8. Figure 12: elements mapping should be moved to supplementary information along with their composition. Only SEM images (at a higher resolution) should remain
9. Regarding the mechanism: does the authors provide any evidence that both metal are forming ions during corrosion.
10. Did the authors measure the hydrogen evolution in time during immersion
Author Response
The authors present a protection material against AZ31B alloy corrosion based on sercitite/epoxy
The work is intresting in the context of developing new materials against corrosion effect.
Comments:
- Abstract: please provide meaning of PEI
Reply: Thank you for your suggestion. The full name of PEI is added into the abstract. See line 21.
- Abstract: more scientific data are required
Reply: Thank you for your suggestion. We have modified the abstract as you suggested. See line 23 in the revised version of manuscript.
- Figures 2 c, d, e: the text (data) from figures should be more clear (a higher resolution or higher text fonts is required)
Reply: Thank you for your suggestion. We have modified Fig 3 for better presentation.
- Figure 2 (SEM) please rename figure 2 with figure 3. Also, figure 3 show a lack of clarity (to many information). I would suggest that the pictures corresponding to the mapping of elements and their table to be moved at supplementary informations
Reply: Thank you for your suggestion. We have modified Fig 3 as you suggested. The maps of elements and their table have been moved to supplementary informations. We have also updated the caption of figure 3 for better clarity. Please see line 274.
- Please provide an explanation for the corrosion potential evolution (for instance why Ecor for Ma alloy is more positively when compared with E, E-S1, and E-S2)
Reply: Thank you for your question. The reasons for the change of corrosion potential include the following two aspects:
- The change of corrosion potential is closely related to the corrosion process of the coated samples. It is well known that the failure process of the coated samples with defects is a typical galvanic cell process. These defects and pores in the epoxy coating can cause the penetration of the solution and the occurrence of local corrosion. When the solution reaches the metal-coating interface, the exposed areas of the AZ31B substrate are severely corroded. At the same time, most of the substrate surface is still covered by the coating. These covered areas and exposed areas form a practical galvanic cell structure, which results in a certain corrosion potential decrease. Therefore, after the addition of highly-orientated SER, the impermeability of the coating was improved, the local corrosion degree was reduced, and its corrosion potential was correspondingly increased. At the same time, the corrosion current was also reduced, and the corrosion resistance was enhanced.
- For the substrate sample AZ31B that we tested, a passivation layer is formed on its surface in air. Although this passivation layer will be destroyed in the corrosion solution, its re-passivation and destruction are two opposite and simultaneous processes. This makes the corrosion potential measured much higher than the standard potential of magnesium (-2.37V). For the samples E, ES1, and ES2 that we tested, after soaking for a period of time, their surfaces have undergone the galvanic cell corrosion process mentioned above. Although the local exposed areas on their surfaces are small, the corrosion tendency of these areas is more serious than the average situation of the whole soaked substrate surface. Therefore, the corrosion potential of E, ES1 and ES2 is slightly lower than the AZ31B substrate. However, for these samples, since the exposed area are extremely small scale area, the corrosion current density calculated by the whole sample area is much smaller than that of the AZ31B sample.
For better clarity, we have added some explaination into the manuscript for better explaination. See line 293.
- Figure 5d: higher resolution of the text is required
Reply: Thank you for your suggestion. We have modified the Fig.5 as you suggested.
- Figure 6 should be moved to supplementary information (small information provided in manuscript with regard to this figure) with a higher resolution
Reply: Thank you for your suggestion. Figure 6 presents original and fitted EIS data of different samples after immersion for different times. We believe this figure can be important and helpful to support the study of long term corrosion behavior. So we prefer to keep this figure. You may try to magnify the picture and see that if the resolution of the picture is good. The journal of Nanomaterials also provide the download of full image picture in the online version of paper.
- Figure 12: elements mapping should be moved to supplementary information along with their composition. Only SEM images (at a higher resolution) should remain
Reply:
Reply: Thank you for your suggestion. We believe that these elemental maps are useful for interpreting the corrosion behavior. Hence, instead of deleting all of them, to enlarge the SEM pictures and improve the resolution of SEM, we deleted the pictures of Cl elements, which are not very informative and helpful. Hope our modification meets your requirement.
- Regarding the mechanism: does the authors provide any evidence that both metal are forming ions during corrosion.
Reply: Thank you for your suggestion. In Fig.10 and Fig.11, we detect the formation of Mg-Al LDH as corrosion product. Mg-Al LDH can only be generated with the participation of aluminum ions.
- Did the authors measure the hydrogen evolution in time during immersion
Reply: Thank you for your question. In our study, we chose electrochemical and immersion tests as corrosion measurements. We did not measure the hydrogen evolution in our experiment.

Reviewer 2 Report
This is an interesting area for study and the reason for the paper is clear.
The paper is well written. The state of art allows us to understand the originality of the paper. The proposed method is detailed and validated with experimental data. However, I feel that there are some points which require improvement in the rigor and detail. I couldn't find an explanation as to why the corrosion potential values change quite randomly (for E, E-S1 and E-S2, it moves to a more negative range), although it is very correctly described in line 298 how it should be. To which diffusion process CPEdiff and Rdiff are assigned? Why does this phenomenon occur only in case of E-S1, E-S2 and E-S6 and not in the case of E-S4? How does this depend on composition (increasing sericite content)? It would be useful to find answers to these questions in the article.
Author Response
This is an interesting area for study and the reason for the paper is clear.
The paper is well written. The state of art allows us to understand the originality of the paper. The proposed method is detailed and validated with experimental data. However, I feel that there are some points which require improvement in the rigor and detail. I couldn't find an explanation as to why the corrosion potential values change quite randomly (for E, E-S1 and E-S2, it moves to a more negative range), although it is very correctly described in line 298 how it should be.
Reply:Thank you for your question. The reasons for the change of corrosion potential include the following two aspects:
- The change of corrosion potential is closely related to the corrosion process of the coated samples. It is well known that the failure process of the coated samples with defects is a typical galvanic cell process. These defects and pores in the epoxy coating can cause the penetration of the solution and the occurrence of local corrosion. When the solution reaches the metal-coating interface, the exposed areas of the AZ31B substrate are severely corroded. At the same time, most of the substrate surface is still covered by the coating. These covered areas and exposed areas form a practical galvanic cell structure, which results in a certain corrosion potential decrease. Therefore, after the addition of highly-orientated SER, the impermeability of the coating was improved, the local corrosion degree was reduced, and its corrosion potential was correspondingly increased. At the same time, the corrosion current was also reduced, and the corrosion resistance was enhanced.
- For the substrate sample AZ31B that we tested, a passivation layer is formed on its surface in air. Although this passivation layer will be destroyed in the corrosion solution, its re-passivation and destruction are two opposite and simultaneous processes. This makes the corrosion potential measured much higher than the standard potential of magnesium (-2.37V). For the samples E, ES1, and ES2 that we tested, after soaking for a period of time, their surfaces have undergone the galvanic cell corrosion process mentioned above. Although the local exposed areas on their surfaces are small, the corrosion tendency of these areas is more serious than the average situation of the whole soaked substrate surface. Therefore, the corrosion potential of E, ES1 and ES2 is slightly lower than the AZ31B substrate. However, for these samples, since the exposed area are extremely small scale area, the corrosion current density calculated by the whole sample area is much smaller than that of the AZ31B sample.
To which diffusion process CPEdiff and Rdiff are assigned? Why does this phenomenon occur only in case of E-S1, E-S2 and E-S6 and not in the case of E-S4? How does this depend on composition (increasing sericite content)?
Reply:Thank you for your question. This can be due to the corrosion behavior as we explained in the last question. These exposed area of ES-1, ES-2, or ES-6 caused by defects possess very high corrosion activity due to galvanic effect. When the corrosion rate is high, the corrosion process is diffusion control. Hence, the resistance and capacitance of the diffusion process appear. In contrast, for ES-4, since the compacity of the coating is good, the corrosion behavior turns from diffusion control to reaction control.
It would be useful to find answers to these questions in the article.
Reply: Thank you for your suggestion. For better clarity, we have added some explaination into the manuscript for better explaination. See line 293.

Reviewer 3 Report
Comments on the paper:
The manuscript titled: "Highly-oriented sericite nanosheets in epoxy coating for excellent corrosion protection of AZ31B Mg alloy" provides new insights into the design of composite epoxy coatings incorporating sericite nanosheets clearly demonstrating their superior anticorrosion performance.
Before publication there are few remarks to consider.
1) In Figure 2, in line 196, in the title of the figure, the (c) is missing
2) In line 260, please correct "...Fig. 3j,n and s,... " to "...Fig. 3i, n and s,... "
3) In line 273, please correct "Figure 2. " to " Figure 3. "
In Discussion, in equation (7) please correct "OH" to "OHˉ
Author Response
The manuscript titled: "Highly-oriented sericite nanosheets in epoxy coating for excellent corrosion protection of AZ31B Mg alloy" provides new insights into the design of composite epoxy coatings incorporating sericite nanosheets clearly demonstrating their superior anticorrosion performance.
Before publication there are few remarks to consider.
1) In Figure 2, in line 196, in the title of the figure, the (c) is missing
Reply: Thank you for your suggestion. We have modified the manuscript as you suggested. Corrections in the revised version have been highlighted.
2) In line 260, please correct "...Fig. 3j,n and s,... " to "...Fig. 3i, n and s,... "
Reply: Thank you for your suggestion. We have modified the manuscript as you suggested. Corrections in the revised version have been highlighted.
3) In line 273, please correct "Figure 2. " to " Figure 3. "
In Discussion, in equation (7) please correct "OH" to "OHˉ
Reply: Thank you for your suggestion. We have modified the manuscript as you suggested. Corrections in the revised version have been highlighted.
